# A 3D-Printed Aqueous Drainage Tube with an Expandable Inner Diameter to Accommodate the Intraocular Pressure (IOP) Fluctuations After Glaucoma Surgery

**DOI:** 10.3390/polym17010118

**Published:** 2025-01-05

**Authors:** Jae-Seok Kim, Hun-Jin Jeong, Ji-Woo Park, So-Jung Gwak, Jeong-Sun Han, Kyoung In Jung, Seung-Jae Lee

**Affiliations:** 1Department of Mechanical Engineering, Wonkwang University, 460 Iksandae-ro, Iksan 54538, Republic of Korea; jaeseokkim311@gmail.com; 2Regenerative Engineering Laboratory, Columbia University, 630W 168th ST, New York, NY 10032, USA; hj2607@cumc.columbia.edu; 3Department of Chemical Engineering, Wonkwang University, 460 Iksandae-ro, Iksan 54538, Republic of Korea; wldn0106@naver.com (J.-W.P.);; 4MECHABIO Group, Wonkwang University, 460 Iksandae-ro, Iksan 54538, Republic of Korea; 5Department of Ophthalmology, Seoul St. Mary’s Hospital, College of Medicine, The Catholic University of Korea, Seoul 06591, Republic of Korea; winehan@catholic.ac.kr (J.-S.H.); ezilean@hanmail.net (K.I.J.); 6Division of Mechanical Engineering, Wonkwang University, 460 Iksandae-ro, Iksan 54538, Republic of Korea

**Keywords:** glaucoma, aqueous drainage tube, 3D printing, tri-axial nozzle, biodegradation

## Abstract

Glaucoma treatment involves reducing the intraocular pressure (IOP), which can damage the optic nerve, to a normal range. Aqueous drainage devices may be used for treatment, and a variety of devices have been proposed. However, they have a non-variable and uniform inner diameter, which makes it difficult to accommodate the IOP fluctuations that occur after glaucoma surgery. To ensure effective treatment in the early post-operative period with low IOP and the late post-operative period with high IOP, the inner diameter should be expandable over time to allow for smooth aqueous drainage. Here, we applied 3D printing technology with a tri-axial nozzle to develop an aqueous drainage tube that can expand its inner diameter. The distinct bilayer structure of the device allows it to expand its inner diameter through biodegradation, which can accommodate the IOP fluctuations that often occur after glaucoma surgery. The fabricated structure was evaluated in a series of tests, including leakage, cytotoxicity, and degradation experiments. The device did not show any leakage, was not toxic to cells, and demonstrated the expansion of the inner diameter through biodegradation. The device may provide a more effective post-operative solution for glaucoma patients by alleviating the effects of low IOP in the early post-operative period and high IOP in the late post-operative period.

## 1. Introduction

Various diseases affect the eye, including cataracts, dry eye, presbyopia, retinal diseases, macular degeneration, and glaucoma. The World Health Organization (WHO) has defined glaucoma, diabetic retinopathy, and macular degeneration as the three leading causes of blindness. In particular, glaucoma is a progressive optic neuropathy and the leading cause of irreversible vision loss [1]. An estimated 70 million people in the world and more than 2 million people in the United States are afflicted with glaucoma. According to OECD statistics, the proportion of people with glaucoma in their 40s and older is almost 84%, and with the increasingly disproportionate number of aging people, more people are expected to be affected by glaucoma in the future [2,3,4,5]. The primary factor causing glaucoma is elevated intraocular pressure (IOP), the pressure within the eye [2,4]. In the normal eye, the ciliary body produces a fluid known as the aqueous humor, and this fluid is drained through a trabecular meshwork, which is a sponge-like tissue [6,7]. The production and drainage of the aqueous humor allow for the maintenance of the IOP in a normal range of 10 to 21 mmHg [8]. However, if the trabecular meshwork becomes damaged or blocked due to inflammation, trauma, or genetic factors, the aqueous humor is not drained properly, and the IOP increases. This can result in pressure on the blood vessels in the eye, causing poor circulation and the compression of the optic nerve, which directly damages retinal cells and the optic nerve [2,9].

The treatment of glaucoma requires IOP control to ensure it remains in a normal range [4,5]. By inhibiting the production of the aqueous humor or promoting its drainage, elevated IOP can be treated, thus alleviating damage to the optic nerve and retinal cells and preventing the progression of visual field defects [1,2,10,11]. Glaucoma treatment is broadly divided into non-surgical and surgical treatments. In the early stages of glaucoma, the first line of treatment is the administration of non-surgical medications [12]. These medications are used to increase blood flow or inhibit the production and drainage of the aqueous humor from the eye. However, if there is an adverse reaction to medications, secondary treatments including a trabeculectomy or laser cyclophotocoagulation are required to create a channel for the aqueous humor to drain [12,13,14]. However, reoperation may be necessary, or complications may occur. In such cases, a drainage device is implanted, which has a relatively low incidence of complications and a high success rate [1,3,14]. Aqueous humor devices reduce the IOP by bypassing the normal aqueous humor pathway and allowing the aqueous humor to drain freely through the drainage channels of the device (Figure 1a).

The Ahmed^®^ valve (New World Medical, Rancho Cucamonga, CA, USA) and the Baerveldt^®^ glaucoma device (Pharmacia, Irvine, CA, USA) are the most common aqueous drainage devices to date, consisting of a plate attached to one end of a silicone drainage tube [15,16]. This structure has the disadvantage of requiring a large incision for insertion due to the relatively large size of the plate. For this reason, minimally invasive glaucoma surgery (MIGS) has attracted interest in recent years, which utilizes a tube-shaped device to minimize the incision [17,18]. Although aqueous drainage devices are inserted to control the IOP, patients will still experience periods of low IOP for one to two weeks after insertion, followed by periods of high IOP for at least one to six months, which can lead to complications and side effects [19]. In the early post-operative period, low IOP can result in complications such as corneal damage, choroidal effusion, and suprachoroidal hemorrhage. Conversely, during the late post-operative period, high IOP can cause fibrous encapsulation around the device, leading to reduced aqueous outflow, device malfunctions, and eventual vision loss due to progressive optic nerve damage. However, the products used in clinical practice are made from a single material and have a non-variable inner diameter. So, these currently used tubes with a constant diameter do not adequately respond to changes in the intraocular pressure (IOP) during the post-operative period. In the early phase, they can cause excessive drainage, leading to hypotony and related complications. As fibrosis progresses over time, the IOP tends to rise, and a constant diameter tube cannot accommodate the increased drainage needs. Consequently, it may not be able to function adequately when the IOP is initially low and subsequently high following glaucoma surgery.

In this study, we used 3D printing technology to improve conventional aqueous drainage devices by fabricating a double-layered aqueous drainage tube using polycaprolactone (PCL) with different molecular weights to expand the inner diameter through biodegradation. In this device, inner diameter expansion is possible due to the use of low-molecular-weight PCL inside the tube. Therefore, it can drain less aqueous humor when the IOP is low during the early post-operative period, and the drainage is increased when the IOP is high during the late post-operative period. The fabricated drainage device could be an attractive alternative to devices currently used in clinical settings (Figure 1b).

## 2. Materials and Methods

### 2.1. Preparation of Polymer Inks

To prepare high-molecular-weight PCL inks for the outer layer, PCL (Mw: 80,000; Sigma-Aldrich, St. Louis, MO, USA) was dissolved in dichloromethane (DCM; purity > 99.5%; Daejung Chemicals and Materials, Siheung, Republic of Korea) for 24 h to obtain a clear solution with the desired concentration. To prepare printable low-molecular-weight PCL inks for the internal layer, PCL (Mw: 10,000; Sigma-Aldrich, St. Louis, MO, USA) and polyethylene oxide (PEO; Mv: 100,000; Sigma-Aldrich, St. Louis, MO, USA) were mixed (2:1 wt%) and dissolved in DCM for 24 h to obtain a clear solution with the desired concentration. To prepare a sacrificial layer ink that could sustain its shape after gelation, Pluronic F-127 (PF-127; Sigma-Aldrich, St. Louis, MO, USA) was dissolved in deionized water (DW), and the solution was refrigerated at 4 °C for 24 h to ensure complete dissolution.

### 2.2. Optimization of Ink Concentration

To confirm the concentration of the PCL ink suitable for printing, an extrusion test was conducted with various concentrations of the ink. High-molecular-weight PCL inks were prepared at concentrations of 20 wt%, 25 wt%, and 30 wt%, and low-molecular-weight PCL inks were prepared at concentrations of 40 wt%, 45 wt%, and 50 wt%. The prepared PCL ink was extruded from the air through a 20 G stainless nozzle at a pressure of 550 kPa for 1 min, and the weight of the polymer ink extruded per minute was measured using an electronic scale.

To evaluate the gelation and shape retention of PF-127, the ink was mixed at concentrations of 25% (*w*/*v*), 30% (*w*/*v*), 35% (*w*/*v*), and 40% (*w*/*v*). The PF-127 ink mixtures were dispensed in 1 mL increments into 5 mL glass vials and kept at 17 °C for 1 h. After 1 h, the glass vials were inverted to observe gelation. In addition, to confirm shape retention after printing, a strand shape was printed on a Petri dish using a 20 G nozzle. After 1 h, the shape retention of the printed strand was examined under an optical microscope.

### 2.3. The Fabrication of the Aqueous Drainage Tube

To fabricate an aqueous drainage tube, a 22 G/17 G/14 G stainless metal tri-axial nozzle (Ramé-Hart Instrument Co., Ledgewood, NJ, USA) was used (Figure 2a). High-molecular-weight PCL ink (Mw: 80,000) and low-molecular-weight PCL ink (Mw: 10,000) were extruded at the outer and middle layers using 560 kPa and 100 kPa, respectively. For the lumen formation, PF-127 ink (40 *w*/*v*%) was extruded with 220 Kpa in the core (Figure 2b). To prevent bubble formation on the wall caused by DCM vaporization, printing was performed at 17 °C. The printed structure was dried in a vacuum oven at 40 °C for 24 h to remove residual DCM and cut into 10 mm segments. Finally, each segment was soaked in DW at 4 °C for 24 h to remove PF-127 from the core part and dried to obtain an aqueous drainage tube (Figure 2c).

### 2.4. Scanning Electron Microscopy Imaging

The diameter and morphological characteristics of the fabricated aqueous drainage tube were analyzed using field emission scanning electron microscopy (FE-SEM; S-4800; Hitachi, Tokyo, Japan). Samples were coated for 2 min at 10 mA for imaging. The captured images were used to measure the diameter with ImageJ (National Institutes of Health, Bethesda, 179 MD, USA).

### 2.5. Experimental Setup for Leakage Test of the Aqueous Drainage Tube

To determine if there was a manufacturing defect, a liquid leakage experiment was performed using a fluorescent solution. The fluorescent solution was prepared by mixing 2 mL of Fluoro-Max Green (a 2 μm particle size; Thermo Scientific, Waltham, MA, USA) in 50 mL of 1X phosphate-buffered saline (PBS). A 22 G needle was sealed with parafilm on both sides of an aqueous drainage tube and connected to a 50 mL syringe containing a fluorescent solution on one side. The connected aqueous drainage tube was dipped into a 1X PBS bath. The flow of the fluorescent solution was set at a rate of 3 mL/min, and the leakage was determined with UV light at a 365 nm wavelength.

### 2.6. Accelerated Degradation Experiment to Confirm Inner Diameter Expansion

To demonstrate that the aqueous drainage tube is capable of expanding the inner diameter as it biodegrades in the human body, an accelerated degradation experiment was conducted using sodium hydroxide (NaOH) [20]. A 5 mm long aqueous drainage tube was prepared and then soaked in a glass vial containing 5 mL of 5 M NaOH solution. Air bubbles were removed using a vacuum desiccator for 1 h. The vial was then stored in an incubator at 37 °C for 2 days. After 2 days, the sample was collected, washed twice with DW, and subsequently dried. Optical microscopy was used to observe changes in the inner diameter and cross-sectional area, which were calculated by ImageJ.

### 2.7. Cell Culture

Primary human Tenon’s fibroblasts (hTFs) were provided by the laboratory of Professor Kyoung In Jung at the Catholic University of Korea, Seoul St. Mary’s Hospital. hTFs were cultured in Dulbecco’s modified Eagle’s medium (DMEM) with high glucose (HyClone, South Logan, UT, USA) supplemented with 10% fetal bovine serum (FBS; Gibco, Carlsbad, CA, USA), 1% antibiotic–antimycotic 100X (Gibco, Carlsbad, CA, USA), and 3% 1 M HEPES buffer (Welgene, Gyeongsan, Republic of Korea) at 37 °C with 5% CO_2_. The culture medium was replaced every 2 days.

### 2.8. In Vitro Cytotoxicity Test

The aqueous drainage tube was evaluated for cytotoxicity. The cultured cells were seeded in 48-well plates at a density of 2 × 10^4^ cells/well and cultured for 1 day. The aqueous drainage tube was cut into 5 mm sections and sterilized by soaking the sections in 70% ethanol overnight, followed by washing them 5 times with sterilized 1X PBS and drying them. The sectioned structures were placed, after changing the medium, in the 48-well plates (*n* = 10). Cell proliferation was evaluated using the Cell Counting Kit-8 (CCK-8; DonginLS, Seoul, Republic of Korea). The absorbance was measured at 1, 3, and 7 days with a microplate spectrophotometer (Epoch™; Winooski, VT, USA). The control group consisted of wells with cultured cells without the drainage tube sample. The cell viability was calculated using the measured absorbance at 1, 3, and 7 days, as shown in Equation (1).
(1)$$Cell\;viability\;(\%)=\left(\frac{{Average\;of\;Absorbance\;}_{Experimental\;group}}{{Average\;of\;Absorbance\;}_{Control\;group}}\right)\times 100$$

### 2.9. Statistical Analysis

The images were analyzed using ImageJ. The statistical analysis of the data was performed using GraphPad Prism 10 (GraphPad Inc., San Diego, CA, USA). Data were expressed as the mean ± the standard deviation. A *t*-test and one-way analysis of variance (ANOVA) were carried out, followed by Tukey’s post hoc test for multiple comparisons between experimental groups. *p*-values of less than 0.05 were considered statistically significant.

## 3. Results

### 3.1. Optimization of PCL Ink and PF-127 Ink Concentrations for 3D Printing

PCL ink has different viscosities depending on the ratio in a DCM solvent. The viscosity affects the extrusion rate of the strands, with low concentrations being unsuitable for 3D printing and high concentrations resulting in low extrusion rates and decreased productivity. Therefore, it was important to determine the optimal concentration of the PCL ink. We assessed the viscosity by preparing PCL inks at various concentrations, using both high- and low-molecular-weight PCL. High-molecular-weight PCL (Mw: 80,000; Sigma-Aldrich, St. Louis, MO, USA) inks did not form a strand shape at concentrations lower than 20 wt%, and at higher concentrations, the strand shape was maintained due to the rapid evaporation of the DCM solvent. At 25 wt% and above, extrusion was possible; however, the extrusion rate was significantly lower even at a pressure of 550 kPa. Therefore, 20 wt% was identified as the optimal concentration (Figure 3a). Low-molecular-weight PCL (Mw: 10,000; Sigma-Aldrich, St. Louis, MO, USA) inks did not form a strand shape at concentrations lower than 40 wt%, and at concentrations above 45 wt%, the extrusion rate was significantly lower than at 40 wt% (Figure 3b).

PF-127 ink was used as a sacrificial material to maintain the shape of the inner lumen during the printing of the core–shell structure. Therefore, it was necessary to select a concentration that could maintain the shape of the PCL shell after printing. In this study, various concentrations of PF-127 ink were evaluated to determine whether the shape was maintained after printing. In the inversion test, after 1 h of curing at 17 °C, the ink at 25% (*w*/*v*) was dripping considerably, whereas the ink at 30% (*w*/*v*) was slightly cured with some dripping. At concentrations of 35% (*w*/*v*) and above, the ink was completely cured and was not dripping (Figure 3c). However, the cured inks at 30% (*w*/*v*) and 35% (*w*/*v*) gradually collapsed after 1 h without retaining the shape of the strand, whereas the ink at 40% (*w*/*v*) retained the initial shape well (Figure 3d). Therefore, 20 wt% high-molecular-weight PCL ink, 40 wt% low-molecular-weight PCL ink, and 40% (*w*/*v*) PF-127 ink were selected as suitable inks for printing.

### 3.2. Morphology Analysis of the Aqueous Drainage Tube Fabricated Using a Tri-Axial Nozzle

The average diameter and morphology of the aqueous drainage tube fabricated using a tri-axial nozzle (22 G/17 G/14 G) were evaluated using FE-SEM and ImageJ. It was confirmed that the PF-127 in the core of the fabricated tube was completely removed during the fabrication process to form a conduit (Appendix A). The high-molecular-weight PCL layer and the low-molecular-weight PCL layer were completely separated to form a bilayer tube (Figure 4a,b). In addition, we also observed surface roughening due to the leaching of PEO from the low-molecular-weight PCL ink during the DW dissolution process to remove PF-127 (Figure 4c). This is expected to have increased the surface area of the low-molecular-weight PCL layer, leading to accelerated degradation. In addition, the fabricated structure had an average outer diameter of 947.91 ± 73.31 μm and an average inner diameter of 661.15 ± 55.56 μm (Figure 4d).

### 3.3. Result of the Leakage Test of the Aqueous Drainage Tube

The aqueous drainage tube manufactured using a tri-axial nozzle was evaluated to confirm that it had been manufactured without defects. The results of the leakage test showed that the micro-sized particles in the fluorescent solution flowed out of the outlet without leakage while passing through the tube. No fluorescent material was observed in the 1X PBS bath during the flow of the fluorescent solution. Therefore, the findings demonstrated that the fabricated structure could allow for the smooth flow of liquid without any defects (Figure 5a,b).

### 3.4. Evaluation of Inner Diameter Expansion Through Biodegradation

An aqueous drainage tube is required to drain a small amount of aqueous humor during the early post-operative period with low IOP after glaucoma surgery, as well as a large amount of aqueous humor during the late post-operative period with high IOP. Therefore, an evaluation of the inner diameter expansion was performed to confirm the functionality of the aqueous drainage tube. Before exposure to 5 M NaOH, the inner diameter was 559.26 ± 18.90 μm, and the cross-sectional area of the low-molecular-weight PCL layer was 0.18 ± 0.02 mm^2^ (Figure 6a). After 2 days of exposure to NaOH at 37 °C, the average diameter was increased to 644.73 ± 25.53 μm, and the average cross-sectional area was decreased to 0.06 ± 0.01 mm^2^ (Figure 6b). The results showed that exposure to NaOH induced the degradation of the low-molecular-weight PCL layer, which led to a decrease in the cross-sectional area and a corresponding increase in the average diameter. Therefore, the diameter could increase with biodegradation in vivo, which may allow for pressure regulation.

### 3.5. The Cell Toxicity of the Aqueous Drainage Tube

When the aqueous drainage tube is implanted in the human body, it should not exhibit toxicity. To evaluate the fabricated structure for toxicity, sectioned samples were placed in 48-well plates seeded with cells, and a cytotoxicity assay was performed using the CCK-8. The cell viability was 96.01 ± 11.68% on day 1, 88.36 ± 10.28% on day 3, and 87.88 ± 10.55% on day 7 (Figure 7). These results demonstrated that the DCM used in the fabrication process was completely removed and not toxic to the cells.

## 4. Discussion

In this study, we developed an aqueous drainage tube with a bilayer structure by 3D printing three different materials simultaneously using a tri-axial nozzle for effective IOP control in the early and late post-operative period. We used PCL as the material for the bilayer shell, which is a synthetic polymer that is non-toxic to cells, biocompatible, biodegradable, and FDA-approved [21]. In addition, the degradation rate of PCL is easy to control by adjusting the molecular weight and additive materials such as polyethylene glycol (PEG) [22]. PCL is a common material used for drug delivery, and it can be loaded with antifibrotic drugs such as mitomycin C and 5-FU [23,24,25,26,27]. These drugs can be released from the aqueous drainage device to prevent the fibrosis-associated blockage of the drainage pathway of the aqueous humor, which can lead to increased IOP [28].

Co-axial nozzles including tri-axial nozzles can be used to print a variety of materials with multiple layers in a single strand or to fabricate tubular structures by removing the core. In addition, when combined with 3D printing technology, the designed shape can be printed precisely, and the shell thickness of the structure can be controlled by adjusting the printing parameters. Due to these advantages, co-axial nozzles and 3D printing technologies have been applied to mimic three-dimensional vascular structures or biological environments [29,30,31,32,33,34,35,36]. In another study, a drug was loaded into the core for gradual release [37,38]. Considering these advantages, designing and fabricating patient-specific shapes, regulating the amount of drugs delivered by adjusting the printing parameters, and multi-drug delivery may be possible in the future.

In this study, the fabricated structure was composed of two completely distinct layers of PCL with an average outer diameter of 947.91 ± 73.31 μm and an average inner diameter of 661.15 ± 55.56 μm, and we observed that the process of soaking it in DW was leaching the PEO, resulting in a rough inner layer [39]. The thickness of the outer layer was measured at 47.49 ± 22.47 μm, and the inner layer was 86.98 ± 42.61 μm (Appendix A). We assume that the large standard deviation is due to the poor alignment of the material to the center using a tri-axial nozzle, resulting in a large variation in the thickness. The cross-sectional area of each layer was confirmed to be relatively reproducible, with values of 0.14 ± 0.03 mm^2^ for the outer layer and 0.23 ± 0.05 mm^2^ for the inner layer (Appendix A). Therefore, future studies should consider how constant thickness control could improve the reproducibility and uniformity of the tubes. A fluorescence leakage test indicated no functional defect. Therefore, the structure could perform the basic function of an aqueous drainage device, which is to drain the aqueous humor. However, the Ahmed^®^ valve and Baerveldt^®^ glaucoma device, which are currently representative drainage devices for clinical use, have a tube outer diameter of 600 μm and inner diameter of 300 μm [4]. Therefore, for effective clinical applications, it is necessary to further reduce the diameter of the currently fabricated devices.

In the accelerated degradation experiment, the cross-sectional area was decreased following the degradation of the internal PCL layer, and the inner diameter was increased from 559.26 ± 18.90 μm to 644.73 ± 25.53 μm. The results suggest that the expansion of the inner diameter by around 32.9% could effectively increase the amount of aqueous humor drained during the late post-operative period with high IOP, leading to an IOP reduction. To further increase the clinical relevance of the findings, studies of PCL degradation during the drainage of the aqueous humor are needed to determine the exact timeline of inner diameter expansion and its association with IOP control over time. Establishing a predictable degradation curve of PCL under physiological conditions could ensure optimized pressure control during the post-operative period.

It is necessary to print a PCL structure using a tri-axial nozzle, which requires a viscous ink. Therefore, PCL was dissolved in DCM to prepare a viscous solution suitable for printing. The printed PCL structure was dried in a vacuum oven at 40 °C for 24 h and then soaked in DW to remove the PF-127 sacrificial layer. Various studies have used synthetic polymers dissolved in a solvent to fabricate 3D structures, followed by the complete removal of the solvent [38,40,41]. If the solvent is not removed appropriately, cytotoxicity may occur due to the presence of the solvent [42]. To evaluate the cytotoxicity, we examined the viability of cells cultured with the fabricated structure using the CCK-8, and there was no significant difference in the proliferation of hTF cells compared to the control at 1, 3, and 7 days. The graph shows a decreasing trend in the cell viability for the experimental group over time, but the optical density (OD) values are increasing, indicating that the cells are still proliferating (Appendix A). The reason the viability appears to be lower is because the cell proliferation rate is relatively low compared to the control group. In addition, after removing the DCM, the bilayer tube made of PCL showed no defects such as cracks; thus, it could function well as an aqueous drainage tube. The findings suggest that the protocol of drying in a vacuum oven at 40 °C and DW washing for DCM removal during the fabrication of the drainage tube could ensure cell compatibility and functionality.

## 5. Conclusions

In this study, high-molecular-weight PCL (for the outer layer) and low-molecular-weight PCL (for the inner layer) were used to fabricate an aqueous drainage tube with an inner diameter that could be expanded via biodegradation. The tube could accommodate the IOP fluctuations over time through biodegradation, thus providing a potential approach to managing complications associated with glaucoma surgery, especially low IOP in the early post-operative period and high IOP in the late post-operative period. Furthermore, the inner layer could be loaded with antifibrotic agents such as mitomycin C or 5-FU, which could be applied to various ophthalmic surgeries requiring IOP control. The use of 3D printing technology has the advantage of producing structures with various geometries. This study primarily aimed to evaluate the feasibility of the concept by demonstrating the potential for inner diameter expansion through degradation. While the results confirm the practicality of the proposed approach, further studies are planned to optimize the degradation rates by controlling the molecular weight and blending ratios of PCL to achieve more precise IOP control. Additionally, research will focus on enhancing the material properties and manufacturing efficiency to ensure the tube’s practical application in clinical settings. Future in vivo experiments are also required to validate the safety and efficacy of the tube.

## Figures and Tables

**Figure 1 polymers-17-00118-f001:**
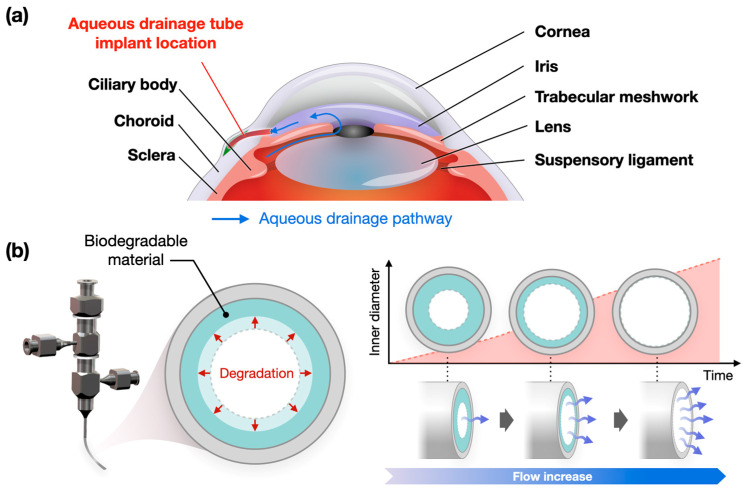
Study concept and design: (**a**) the anatomy of the eyeball (red indicates location of the implanted aqueous drainage tube, blue arrows indicate the aqueous humor); (**b**) a conceptual image of an aqueous drainage tube with an expandable inner diameter (red indicates inner diameter expansion by biodegradation, blue arrows indicate the aqueous humor).

**Figure 2 polymers-17-00118-f002:**
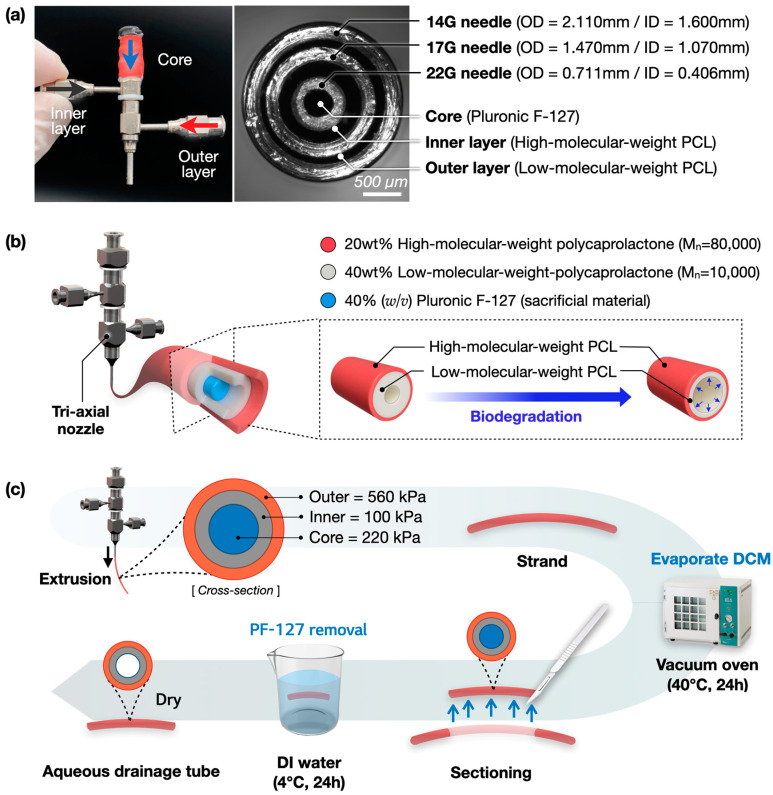
Fabrication of aqueous drainage tube: (**a**) tri-axial nozzle for biomaterial extrusion; (**b**) schematic of biomaterial configuration with different concentrations of PCL and PF-127; (**c**) overall fabrication process for fabricating aqueous drainage tube.

**Figure 3 polymers-17-00118-f003:**
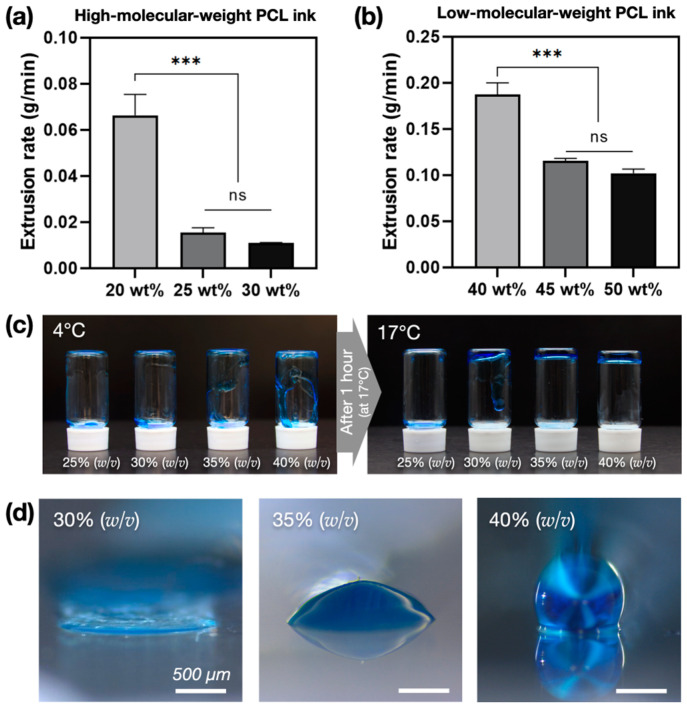
Testing of polycaprolactone (PCL) and PF-127 inks for printability: (**a**) extrusion rate of high-molecular-weight PCL inks; (**b**) extrusion rate of low-molecular-weight PCL inks; (**c**) gelation of PF-127 ink; (**d**) shape retention of PF-127 ink (*** = *p* ≤ 0.001; ns = no significant difference, *p* > 0.05).

**Figure 4 polymers-17-00118-f004:**
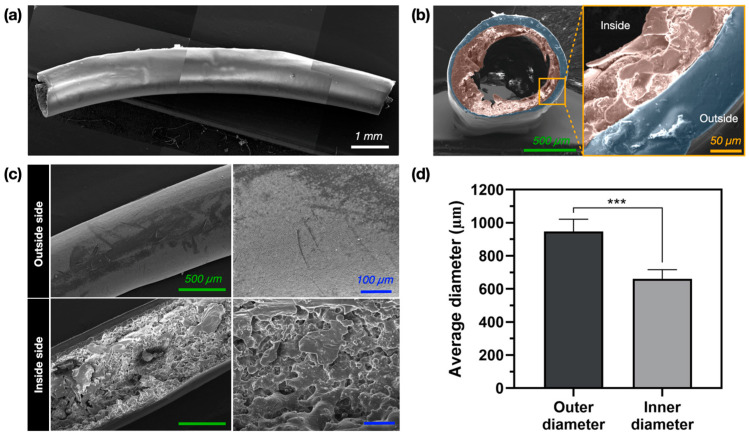
Evaluation of the fabricated aqueous drainage tube: (**a**) overall scanning electron microscopy (SEM) image; (**b**) SEM image of a circumferential cross-section (red indicates the inner layer, blue indicates the outer layer); (**c**) SEM image of a longitudinal cross-section; (**d**) measurements of the inner and outer diameter (*** = *p* ≤ 0.001).

**Figure 5 polymers-17-00118-f005:**
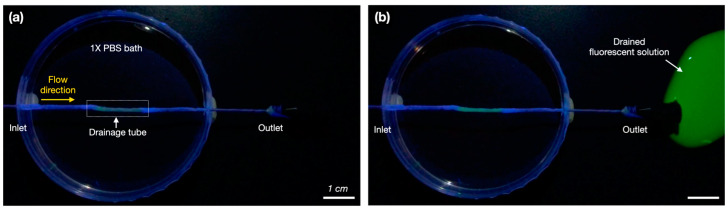
Leakage test of the fabricated aqueous drainage tube: (**a**) Leakage experiment configuration. (**b**) Flow of the fluorescent solution out of the outlet without leakage from the drainage tube.

**Figure 6 polymers-17-00118-f006:**
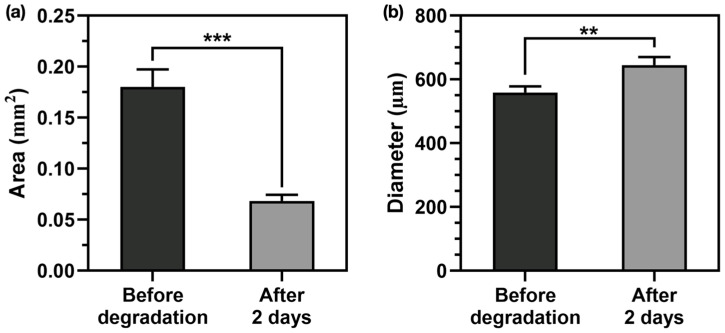
Evaluation of the diameter expansion of the aqueous drainage tube exposed to NaOH: (**a**) measurement of the cross-sectional area of the low-molecular-weight polycaprolactone (PCL) layer before and after degradation; (**b**) measurement of the inner diameter before and after degradation. (*** = *p* ≤ 0.001; ** = *p* ≤ 0.01).

**Figure 7 polymers-17-00118-f007:**
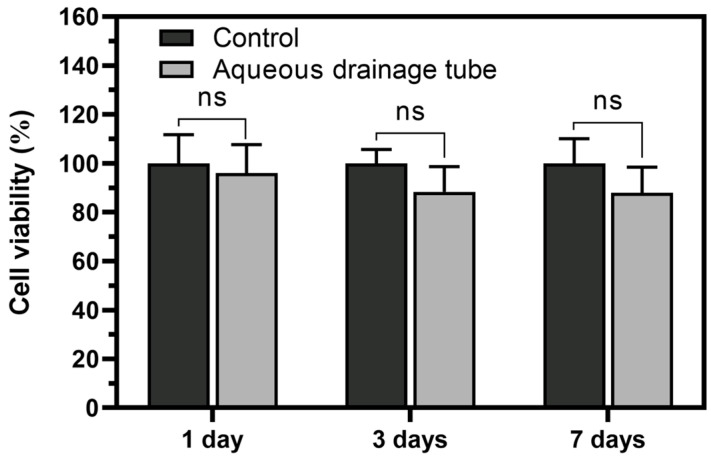
Cytotoxicity evaluation of the fabricated aqueous drainage tube at 1, 3, and 7 days (ns = no significant difference, *p* > 0.05).

## Data Availability

The data are contained within the article.

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
