# Peer review of "A 3D-Printed Aqueous Drainage Tube with an Expandable Inner Diameter to Accommodate the Intraocular Pressure (IOP) Fluctuations After Glaucoma Surgery"

_polymers, 2025, doi:10.3390/polym17010118_

Round 1
Reviewer 1 Report
Comments and Suggestions for Authors
This manuscript need to be revised with the writing criterion.
The figure in introduction should be added with the experimental research part. Please do not use others' work as your experimental research work. It is infeasible.
Meanwhile, it is discourteous for separating the result and the discussion. Why did you submit an artical with the separated result and discussion? Just for causing confusion for the readers?
An unclear reseach process have been presented for current version.
3Dprinting process?
Clear printed structure?
Reviewer 2 Report
Comments and Suggestions for Authors
3D printed aqueous drainage device – It is good drug device combination option for IOP.
Only challenge in present study is as below.
Author mention use of 3D printing technology with a tri-axial nozzle. Any data on validation of 3D printer with a tri-axial nozzle. Layer thickness, coating efficacy and uniformity. Since reproducibility in such cases is quite tedious.
Reviewer 3 Report
Comments and Suggestions for Authors
This manuscript described the use of Bioprinter in coupled with tri-axial nozzle to print a a double-layered aqueous drainage tube by using low and high Mw PCL. Microstructure, leakage and degradations of the samples were determined. Overall, the content should be of interest, but some revisions and addition results are needed to improve the quality and clarity of the manuscript as follows:
-The term "aqueous drainage tube" should be used throughout the manuscript instead of "aqueous drainage device" or "functional aqueous drainage tube" in both title and context for accuracy and consistency.
- Title: Should be changed to accurately reflect the content of the manuscript. This is just a tube, not a device. The term "variable inner diameter" is quite ambiguous and it does not adapt to the IOP after glaucoma surgery. it just changes.
Introduction: The complications and side effects of using the current aqueous drainage device should be given in this section, not to just cite the reference.
-The rationale of the study was still unclear. Was there a problem of the currently used tube with a constant diameter? Why do we should have a tube that the inner diameter can be increased with times? Why cannot we use a tube that has a large diameter enough for drainage? What was the problem if we use large diameter tube with low IOP at the initial postop?
-Method and fig 2: "a 22 G/ 17 G/ 14 G stainless metal tri-axial nozzle". The inner diameter of the needle should also be given for better understanding. Also why these configuration? Since authors already know the dimension of current tube in commercial device, why not using the configuration to produce such dimension or diameter?
-Method: "the segment was soaked in DW at 4℃ to remove PF-127 from the core part and dried to obtain a functional aqueous drainage tube" How long? Should be given and how do authors know that it is enough?
- Method: The shape, dimension and weight of the sample used for the degradation test should be given.
-Method: " 2 × 104 cells/well" Superscipt is needed here.
-Cell study: The detail and description was quite mixed up and unclear. The description of cytotoxicity and proliferation test should be described separately for the testing details. Also the calculation of cell viability and proliferation rate. The control samples should also be given.
-Statistical analysis : "The images were analyzed using ImageJ (National Institutes of Health, Bethesda, 179 MD, USA). Statistical analysis of the data was performed...". The details of ImageJ should be moved to the first description of using this software, not here. Also, the posthoc test used after ANOVA should be mentioned.
- After fabrication, the obtained thickness of both layers should be given and compared to the configuration of the nozzle to see the accuracy of such printed tube.
- Since the idea of using two MW PCL to manufacture a tube will involve the different rate of degradation, but both PCL will degrade anyway during use. How do we know that the tube will not leak after certain periods for upto 12 months due to degradation. At least, leak test should also be performed after degradation test at several time points to see the integrity of the tube.
-Apart from the change in diameter, the morphology and microstructure by SEM after degradation test should be performed and reported to see if there are any changes which could affect the the performance of the samples.
-Figure 7: Statistical analysis should be performed to see if there is any significant difference in cytotoxicity among times and between control or the proliferation rate.
-Discussion:
1. The initial part could be deleted since it duplicated the Introduction and not necessary to be here.
2. The discussion on the morphology and microstructure of the obtained tube should be discussed here whether this is satisfactory or not according to the design. Also, Why outer surface is smooth, but the inner surface is rough? What are the causes? Why the leaching of PEO caused such a feature? Any proof or citation?
3. Although cytotoxicity tests revealed the non-cytotoxicity potential of the fabricated sample for up to 72 h. However, there seemed to be a decreasing trend of the cell viability with times. Discussion of this observation and the causes is needed here. In addition, the proliferation rate increased with times in contrast to the viability. Again, discussion of this observation and the causes is needed here.
- Limitation and future perspective of the studies performed should also be given.
- Conclusion is not a shortened results description. This is where the authors conclude the findings according to the objective of the study. Future works could also be given here.
Round 2
Reviewer 3 Report
Comments and Suggestions for Authors
Authors have revised the manuscript according to the suggestions. However, some points are needed to revise further.
- Title: Although authors have revised the title to “3D printed aqueous drainage tube with an expandable inner diameter that adapts to intraocular pressure (IOP) fluctuations after glaucoma surgery” I would think that the term "adapts" is quite misleading as suggested last time. As noted, the diameter does not adapt, but only increased uncontrollably with times. Title may be changed to for example "3D printed aqueous drainage tube with an expandable inner diameter to accommodate the intraocular pressure (IOP) fluctuations after glaucoma surgery"
-As suggested previously, the term "functional aqueous drainage tube" should be changed to just " Aqueous drainage tube. No change is noted for such term i.e in the context, the sample label in figures i.e Fig. 7, Fig. S3.
- The first part of conclusion could be deleted since it duplicated the results and discussion.
